# Diabetic retinopathy among type 2 diabetes mellitus patients in Sabah primary health clinics–*Addressing the underlying factors*

**Nurul Athirah Naserrudin**[1,2◉]*, **Mohammad Saffree Jeffree**[3◉], **Nirmal Kaur**[2◉], **Syed Sharizman Syed Abdul Rahim**[3◉], **Mohd Yusof Ibrahim**[3◉]

**1** Department of Community Health, Faculty of Medicine, National University of Malaysia, Cheras, Kuala Lumpur, Malaysia, **2** Sabah State Health Department, Malaysia Ministry of Health, Sabah, Malaysia, **3** Faculty of Medicine and Health Science, Universiti Malaysia Sabah, Kota Kinabalu Sabah, Malaysia

◉ These authors contributed equally to this work.
* drathirah85@gmail.com

**Data Availability Statement:** Data cannot be shared publicly because of the confidentiality of the patient. Data are available from the Sabah State

## Abstract

Every person diagnosed with diabetes mellitus (T2DM) is at risk of developing Diabetic retinopathy (DR). Thus, DR is one of the major chronic microvascular complications of T2DM. However, in Malaysia, research about DR is still scarce. This study aimed to determine the prevalence of DR among diabetic patients across 46 primary healthcare clinics in Sabah, Malaysia. Secondly, it purported to identify the factors influencing the development of DR. This cross-sectional study involved a total of 22,345 Type 2 diabetes mellitus (T2DM) patients in the Sabah Diabetic Registry from 2008 to 2015. Of the 22,345 T2DM patients, 13.5% (n = 3,029) of them were diagnosed with DR. Multiple logistic regression revealed seven major risk factors of DR, i.e. patients with diabetic foot ulcer [aOR: 95% CI 3.08 (1.96–4.85)], patients with diabetic nephropathy [aOR: 95% CI 2.47 (2.13–2.86)], hypertension [aOR: 95% CI 1.63 (1.43–1.87)], dyslipidaemia [aOR: 95% CI 1.30 (1.17–1.44)], glycated haemoglobin [(HbA1c) > 6.5 (aOR: 95% CI 1.25 (1.14–1.38)], duration of diabetes mellitus (T2DM) [aOR: 95% CI 1.06 (1.05–1.07)] and age of patient [aOR: 95% CI 1.01 (1.00–1.02)] respectively. DR is a preventable complication. The effective glycaemic control is crucial in preventing DR. In minimizing the prevalence of DR, the healthcare authorities should institute programmes to induce awareness on the management of DR's risk factors among patient and practitioner.

## Introduction

Type 2 Diabetes mellitus (T2DM) is one of the major non-communicable diseases worldwide. The global incidence of T2DM in 2017 was as high as 22.9 million and this number is expected to rise if effective prevention methods are not put in place [1]. In addition, the data in 2015 highlighted that 8.8% of the adults in Southeast Asia had T2DM. In Malaysia, the prevalence of T2DM was 0.94 million in 2000 and the number is expected to increase by 164.0% to 2.48 million in 2030 [2]. In Peninsular Malaysia, the risk of Diabetic Retinopathy (DR) was higher

Health Department. The data could be accessed by writing a letter or email to the Head of Sabah Health State (Dr Rose Nani Mudin) and posted to Tingkat 3, Rumah Persekutuan, Jalan Mat Salleh, Peti Surat 11290, Sabah, 88590 Kota Kinabalu or email to drrose@moh.gov.my.

**Funding:** The author(s) received no specific funding for this work.

**Competing interests:** The authors have declared that no competing interests exist.

among Malays compared to other races [3]. Diabetes mellitus (DM) is characterised by a high glucose level in the blood (hyperglycaemia) resulting from defective mechanisms in insulin secretion, insulin action, or both [4]. The management of glucose levels among diabetic patients is crucial as poor control of T2DM can affect the patients' life significantly. Furthermore, uncontrolled blood sugar levels may lead to a greater risk of diabetic complications. The three main categories of diabetic complications are microvascular diseases, macrovascular diseases, and immune dysfunction [4]. Specifically, nephropathy, retinopathy, neuropathy diabetic ulcer disease, cardiovascular disease, cerebrovascular incidents are among the commonest complications that result from poor control of T2DM [5, 6].

Among these complications, DR is the major chronic microvascular complication. The prevalence of DR was reported to be 35.0% across 93 million individuals [7]. There are two types of DR, namely non-proliferative and proliferative. Non-proliferative DR, also known as the early stage of DR, can be categorised as mild, moderate, or severe. This stage represents the best window to undertake the necessary intervention to improve diabetes control before further deterioration into reach the stage of proliferative DR. Proliferative DR is diagnosed when there is a presence of neovascularisation [8]. In general, proper categorisation of DR is crucial in determining the right intervention for the patient. Without the right intervention, DR can lead to decreased visual acuity and subsequently blindness [9]. Existing research indicated that the contributing factors of DR included uncontrolled T2DM, longstanding T2DM, the presence of other diabetic complications such as foot ulcer, diabetic nephropathy, hypertension, dyslipidaemia, as well as high level of glycated haemoglobin [5, 6, 10].

DR is one of the major causes of preventable blindness worldwide. A recent study reported that 1 in 10 T2DM patients will develop vision-threatening type of retinopathy that can eventually lead to blindness [11]. Statistics showed that approximately 35.0% of people living with diabetes have developed some degree of DR even at diagnosis [11]. Additionally, DR can also develop after a certain period of disease progression. About 60.0% of individuals will be diagnosed with DR after 20 years of suffering with T2DM [10]. Another study reported that post 15 years of T2DM diagnosis, 2.0% of patients will become blind and 10% will develop some degree of visual impairment due to DR [12]. According to a recent study, DR is the leading cause of preventable blindness among adults between 24 and 74 years old [9]. By correctly diagnosing the various development stages of DR among diabetic patients, the necessary management plan can be provided for the patients at the earliest stage to reduce the prevalence of blindness [13, 14].

In Malaysia, the latest statistics showed that 36.8% of diabetic patients suffered from DR [15]. On a similar note, DR is also one of major causes of blindness in Malaysia, whereby it was reported as the contributing factor of blindness among 10.4% of the elderly patients [16]. The National Diabetic Registry (NDR) was launched in 2006. Recent statistics published in 2017 indicated that the prevalence of DR was highest in Kedah (25.4%) and lowest in Sabah and Wilayah Persekutuan Labuan (14.2%) [17]. The lowest prevalence of DR identified in Sabah could be due attributed to the low levels of health utilisation and diabetes care provided to the community. Throughout the entire Sabah, there is only three fundoscopy cameras to cater for the whole population at Panampang Public Health Clinic, Sandakan Public Health Clinic, and Luyang Public Health Clinic [17]. The lack of fundoscopy camera in other areas of Sabah could have resulted in an underestimation of the prevalence of DR in Sabah. Similar issue on the lack of screening equipment leading to a falsely low prevalence of DR is also experienced in other low-income settings [18].

The low prevalence of DR among the people of Sabah can be linked to the health inequalities and low healthcare utilisation in the area. These issues might arise due to the characteristics of local ecosystem and geographical limitations in remote areas. On top of limited access

to healthcare, poor infrastructures of healthcare facilities in in some areas further complicated the situation. In comparison to Peninsular Malaysia, the socioeconomic progress in Sabah has been relatively slow, mostly due to the vast isolated areas that are inaccessible by roads. Even though public healthcare facilities are available, it is still not feasible for all in the community to receive the necessary treatment due to the lack of infrastructures for them to access the services [19]. These underlying differences may have contributed to the different prevalence of DR in Sabah.

Currently, there is a lack of scientific research on DR in Sabah. Information about the associated risk factors of T2DM complications and comorbidity remains limited. Such information is essential to formulate effective interventions in delaying the progress of DR, subsequently improving the quality of life of diabetic patients. With timely intervention and early detection of the patient's condition, the severity level of complications arising from DR can be kept at a minimum. Therefore, this study aimed to determine the prevalence of DR among patients attending the primary health clinics in Sabah and to identify the risk factor of DR among them.

## Methods

### Study design and population

This was a cross-sectional study to determine the prevalence of DR and its risk factors among 22,345 T2DM patients in Sabah. The clinical and demographic data of the patients registered in the Sabah Diabetic Registry from 2008 to 2015 were obtained. The Sabah Diabetic Registry is a database of diabetic patients registered across 46 primary health clinics in Sabah. The data were entered into the registry by trained healthcare staff in respective clinics. This study only focused on T2DM patients (n = 22,345) in the registry. Hence, patients with type T2DM, impaired fasting glucose, and other types of diabetes such as gestational diabetes were excluded. This study was approved by the National Medical Research Ethical Committee (registered as ID 16-1353-31801). As this study utilised only retrospective data, the ethical committee waived the requirement for informed consent.

### Variables and operational definition

The dependent variable for this study was DR. In this study, DR was defined as the state when there was a disturbance in the visual acuity of diabetic patients and pathological changes were detected seen via fundoscopy using the fundus camera. During data collection, patients' history of DR was screened and classified as "DR present", "DR absent", or "DR unknown". The DR in the registry was not categorised as proliferative or non-proliferative.

Independent variables that were collected included sociodemographic data such as gender, age, duration of T2DM diagnosis, health indices namely HbA1c, blood pressure level, lipid level, BMI, other comorbidities and complications such as nephropathy, cerebrovascular disease, ischaemic heart disease, and diabetic foot ulcer (DFU).

### Analyses

The prevalence of DR was calculated using the number of patients with DR as the numerator and the total number of T2DM as the denominator. Before the factors influencing DR were determined using the Multiple Logistic Regression (MGR), bivariate analysis using the t-test and chi-square were performed to identify the significant factors. Variables that were analysed to identify the risk factors are age, duration of T2DM, HbA1c, nephropathy, DFU,

hypertension, and dyslipidaemia. All data were analysed using IBM SPSS version 20 and the p-value was set at p<0.05.

## Results

Of the 22,345 T2DM patients, 13.5% (n = 3,029) of them were diagnosed with DR. As seen in Table 1, the prevalence of DR was higher among patients above 80 years old, males, patients with hypertension, dyslipidaemia, nephropathy, ischaemic heart disease, cerebrovascular disease, and DFU. The prevalence of DR was also higher among patients with a disease duration of 25 years and above.

The bivariate analysis indicated that patients 80 years old and above were almost five times more likely to develop DR compared to other age groups (CI 95%: 1.62–12.76). The odds ratio of male patients developing DR was 1.1 times higher than females (95% CI: 1.04–1.22). In terms of comorbidities, T2DM patients with hypertension were 2.1 times more likely to get DR compared to the patients without hypertension (95% CI: 1.9–2.4). In addition, the risk of patients with dyslipidaemia developing DR was 1.7 times greater than patients without dyslipidaemia (95% CI: 1.5–1.8).

With regard to T2DM complications, patients with certain conditions showed a higher risk to develop DR, i.e. nephropathy [OR 3.2 (95% CI 2.8–3.7)], ischaemic heart disease [OR 1.7 (95% CI 1.29–2.19)], cerebrovascular disease [OR 1.5 (95% CI 1.05–2.19)], and DFU [OR 4.6 (95% CI 3.1–6.7)]. In addition, DR was also proven to be associated with the duration of the disease. Those with T2DM of 25 years and above were five times more likely of getting DR compared to those with a duration of fewer than 25 years. The findings also highlighted patients with an HbA1c of 6.5% and above had a 1.3 times higher risk of getting diabetic retinopathy (95% CI 1.21–1.43).

The output from MGR revealed seven major risk factors of DR (Table 2). The strongest risk factor were patients with a DFU. The odds ratio for patients with DFU to get DR was three times higher than other patients (CI: 1.96–4.85). Subsequently, the probability of a patient with diabetic nephropathy to get DR was two times higher than other patients (CI: 2.13–2.86). In addition, patients with hypertension were 1.6 times higher to get DR than patients without hypertension (CI: 1.43–1.87). From the analysis, it was revealed that DR is 1.3 times more likely to occur among patients with dyslipidaemia (CI: 1.17–1.44). Also, the incidence of DR is 1.3 times more likely to be diagnosed among patients with HbA1c > 6.5 (CI: 1.14–1.38).

## Discussion

Based on the study, the prevalence of DR among T2DM patients in Sabah was 13.5%. The prevalence was slightly higher than another Malaysian study in Selangor (9.0%) [3]. The higher prevalence reported in this study could be due to the difference in the selection of study sample. In the Selangor study, only patients aged 40 years old and above were included. On the other hand, we included patients below 40 years old as well. In other words, a bigger sample size would have captured a higher population of diabetic patients who have undergone DR screened, thus increasing the prevalence of DR.

From the findings of this study, age is one of the significant risk factors of DR. With increasing age, the risk of developing DR becomes higher among T2DM patients. This finding is supported by previous studies on DR conducted in Singapore. Age was reported to be associated with an increased risk of developing DR with an OR of 2.2 [20]. In another study conducted in the United States, the Wisconsin Epidemiologic Study of Diabetic retinopathy III showed that the severity of DR was associated with younger age at diagnosis [21]. This finding also echoed a Malaysian study in a teaching hospital whereby the risk of developing DR was

**Table 1. Distribution of patients with diabetic retinopathy by risk factors.**

| Variables | Status of retinopathy | | T/Chi-Square Value | P-value | Odds Ratio (95% CI) |
|---|---|---|---|---|---|
| | Yes (%) | No (%) | | | |
| **Age** | | | **t = 13.35** | **<0.05** | N/A |
| < 30 | 4 (7.3%) | 51 (92.7%) | | N/A | N/A |
| 30–39 | 58 (10.9%) | 474 (89.1%) | | 0.41 | 1.6 (0.54–4.47) |
| 40–49 | 278 (13.5%) | 1782 (86.5%) | | 0.19 | 2.0 (0.71–5.55) |
| 50–59 | 838 (16.3%) | 4314 (83.7%) | | 0.08 | 2.5 (0.89–6.87) |
| 60–69 | 1040 (19.6%) | 4261 (80.4%) | | 0.03 | 3.1 (1.12–8.63) |
| 70–79 | 637 (24.5%) | 1958 (75.5%) | | 0.01 | 4.2 (1.49–11.52) |
| > = 80 | 174 (26.3%) | 488 (73.7%) | | 0.01 | 4.6 (1.62–12.76) |
| **Gender** | | | $X^2 = 8.21$ | **<0.05** | **1.1 (1.04–1.22)** |
| Male | 1321 (19.6%) | 5434 (80.4%) | | | |
| Female | 1708 (17.8%) | 7894 (82.2%) | | | |
| **Hypertension** | | | $X^2 = 165$ | **<0.05** | **2.1 (1.9–2.4)** |
| Yes | 2625 (20.6%) | 10097 (79.4%) | | | |
| No | 365 (10.9%) | 2983 (89.1%) | | | |
| **Dyslipidaemia** | | | $X^2 = 118$ | **<0.05** | **1.7 (1.5–1.8)** |
| Yes | 2284 (21.0%) | 8615 (79.0%) | | | |
| No | 685 (13.7%) | 4315 (86.3%) | | | |
| **BMI** | | | $X^2 = 44.76$ | **<0.05** | N/A |
| <25.0 | 1081 (21.6%) | 3932 (78.4%) | | | N/A |
| 25.0–29.9 (overweight) | 1174 (18.6%) | 5146 (81.4%) | | | 0.8 (0.76–0.91) |
| > = 30.0 (obese) | 678 (16.1%) | 3526 (83.9%) | | | 0.7 (0.63–0.78) |
| **Nephropathy** | | | $X^2 = 3.35$ | **<0.05** | **3.2 (2.8–3.7)** |
| Present | 405 (39.3%) | 626 (60.7%) | | | |
| Absent | 2516 (16.6%) | 12632 (83.4%) | | | |
| **Ischaemic heart disease** | | | $X^2 = 14.9$ | **<0.05** | **1.7 (1.29–2.19)** |
| Present | 76 (26.7%) | 209 (73.3%) | | | |
| Absent | 2820 (17.8%) | 13030 (82.2%) | | | |
| **Cerebrovascular Disease** | | | $X^2 = 4.9$ | **<0.05** | **1.5 (1.05–2.19)** |
| Present | 38 (24.8%) | 115 (75.2%) | | | |
| Absent | 2859 (17.9%) | 13117 (82.1%) | | | |
| **Diabetic Foot Ulcer (DFU)** | | | $X^2 = 74.1$ | **<0.05** | **4.6 (3.1–6.7)** |
| Present | 54 (50.0%) | 54 (50.0%) | | | |
| Absent | 2897 (18.0%) | 13238 (82.0%) | | | |
| **Duration of T2DM (Years)** | | | **t = 22.34** | **<0.05** | N/A |
| 0 to 4 | 549 (11.0%) | 4434 (89.0%) | | N/A | N/A |
| 5 to 9 | 1348 (18.4%) | 5976 (81.6%) | | <0.05 | 1.8 (1.64–2.03) |
| 10 to 14 | 684 (24.9%) | 2060 (75.1%) | | <0.05 | 2.7 (2.37–3.03) |
| 15 to 19 | 240 (33.7%) | 473 (66.3%) | | <0.05 | 4.1 (3.45–4.90) |
| 20 to 24 | 96 (31.1%) | 213 (68.9%) | | <0.05 | 3.6 (2.82–4.71) |
| 25 and above | 112 (39.4%) | 172 (60.6%) | | <0.05 | 5.3 (4.08–6.78) |
| **HbA1c** | | | $X^2 = 40.2$ | **<0.05** | **1.3 (1.21–1.43)** |
| > 6.5% | 1606 (21.0%) | 6047 (79.0%) | | | |
| < 6.5% | 1122 (16.8%) | 5547 (53.2%) | | | |

**Table 2. Multivariate analysis of diabetic retinopathy among T2DM patients registered in the Sabah Diabetes Registry.**

| Variables | Beta | Standard Error (SE) | Wald | df | Average Odd Ration (aOR) (95% CI) |
|---|---|---|---|---|---|
| Age | 0.011 | 0.002 | 26.1 | 1 | 1.1 (1.00–1.02) |
| Duration of T2DM | 0.06 | 0.004 | 205.9 | 1 | 1.1 (1.05–1.07) |
| HbA1c (>6.5) | 0.227 | 0.047 | 23.3 | 1 | 1.3 (1.14–1.38) |
| Nephropathy | 0.903 | 0.075 | 143.2 | 1 | 2.5 (2.13–2.86) |
| Diabetic Foot Ulcer | 1.126 | 2.231 | 23.7 | 1 | 3.1 (1.96–4.85) |
| Hypertension | 0.491 | 0.068 | 52.1 | 1 | 1.6 (1.43–1.87) |
| Dyslipidaemia | 0.263 | 0.053 | 24.5 | 1 | 1.3 (1.17–1.44) |
| Constants | -3.457 | 0.146 | 561.7 | 1 | |

higher among the older population due to the vascular changes in the retinal circulation [22]. With advancing age, DR can result in blindness in the elderly if left untreated.

Apart from that, our study also showed that the duration of T2DM diagnosis was significantly associated with DR. An increase in the duration of T2DM diagnosis would also increase the probability of getting DR. This was consistent with similar published research but a higher OR was observed in the other studies [11, 22, 23]. A lower OR in this study could be attributed to the different sociodemographic and geographical characteristics of our study population. In another Malaysian study, 21.0% of the study samples were diagnosed with DR, in which 20.0% of the samples were diagnosed with less than five years of T2DM, 26.0% of them were diagnosed with T2DM between five to ten years, and another 12.0% had been diagnosed for more than ten years [24]. However, a higher prevalence of DR of 21.0% in previous study compared to 13.5% in this study could be due to the broader inclusion criteria of the study population [24].

Furthermore, the development of DR is highly dependent on diabetes management. Effective management of diabetes can ensure good control of the HbA1c level. In contrast, poor management of diabetes may lead to unsatisfactory HbA1c levels that subsequently increase the probability of developing DR. Similar to previous studies, this study also revealed a strong association between HbA1c and the development of DR [11, 23]. Specifically, a 1.0% reduction of HbA1c was associated with a 37.0% reduction in microvascular complications [25]. However HbA1c level is inconclusive and maybe an insignificant finding to DR [22]. Additionally, the lifetime exposure to high blood glucose levels was the principal determinant of DR. Evidence from this study indicated the importance of early intervention in controlling blood sugar levels to lower the risk of DR. In other words, an intense glycaemic control is protective against DR [26]. It can be achieved via strict diet control, healthy lifestyle, physical activity, and low-stress levels. Besides lifestyle modifications and pharmacological therapy, more intensive non-pharmacological management such as pancreas transplant and islet cell transplantation is available for DR at present [27]. Islet cell transplantation has shown promising results in slowing the progression of DR compared to medical management. However, in the abovementioned study, the patients who received the transplant had better baseline HbA1c control than the non-transplanted group [28]. Thus, more studies are warranted to obtain more established evidence on the effect of islet cell transplantation on DR [27]. In terms of medical management, the widespread availability of continuous insulin pump therapy has been associated with greater reduction of DR rate and superior benefits over conventional insulin therapy [29]. The medical team and patients must work closely to keep diabetes under control. In the long term, the benefits of intensive blood sugar control with a lower variation in the glucose level

can halt the progress of DR and other DM complications such as diabetes nephropathy, peripheral vascular disease, and coronary heart disease [29].

Moreover, this study highlighted the association between DR and diabetic nephropathy. This was aligned with the Asian Korean study that reported a 19.3% prevalence rate of DR among patients with nephropathy [30]. The pathophysiology of both DR and diabetic nephropathy is similar. The complications occur following the exposure of the cell membrane to oxidative stress caused by uncontrolled blood sugar levels. In addition, many mechanistic pathways are underlying diabetic complications, including glycation of proteins, and the formation of advanced glycation end products (AGEs). A study on the pathogenesis of retinopathy and nephropathy reported the roles of glycated proteins and AGEs that were associated with prior HbA1c levels as the basis of the diabetic phenomenon of "metabolic memory" in the pathogenesis of diabetes complications [31]. In advanced retinopathy when structural changes are detected in the cell membrane of the eye, similar changes will also be seen in the kidney cells [20, 21].

Worldwide, there are limited studies on DR and its associations with the DFU. However, this study found a strong association between DFU and DR. In a Western study on clinical risk factors and outcomes of DFU, DR was found in 54.0% of DFU patients. The study also reported that the presence of diabetic nephropathy was 26.0% among the patients with DFU [22]. Both DR and DFU are chronic microvascular complications of T2DM as a result of chronic longstanding damage to the blood vessels due to exposure to high blood glucose levels in the blood. These conditions lead to the thickened microvascular basement membrane that results in impaired diffusion of oxygen and nutrients into the cells. Consequently, fewer nutrients are received by the cells. To compensate, neovascularisation will occur and these abnormal blood vessels are prone to rupture and microaneurysms. The pathophysiological mechanisms of these two complications are closely linked to the "metabolic memory", a phenomenon associated with progressive years of diabetes mellitus. The changes in the glycaemic levels of diabetic patients could predispose them to not just retinopathy and DFU, but also other chronic complications simultaneously. DFU can be diagnosed visually and thus, its presence can indicate possible retinopathy changes in diabetic patients [22].

In this study, a strong association was identified between hypertension and the severity of retinopathy. Studies done among the Asian population showed twice the risk of developing retinopathy among diabetic patients with hypertension [32, 33]. Similar findings were also reported in Western countries [10, 11]. Hypertensive patients suffer from abnormal retinal autoregulation, thus making them unable to protect the changes in blood pressure due to the hyperglycaemia in DM patients that impairs the regulation of retinal perfusion [34].

Despite the established clinical relationship, the role of dyslipidaemia in the development of DR has not yet been studied in detail. In Western countries, retinopathy was significantly associated with the higher plasma level of low-density lipoprotein (LDL) cholesterol, but not related with high-density lipoprotein (HDL) or plasma triglyceride levels [35, 36]. On a similar note, a study in Japan reported that the serum lipid concentrations are higher in T2DM patients compared to the normal population, thus likely attributing to the development of DR. In short, T2DM patients with high serum lipids are likely to develop DR, especially the proliferative type [37].

DR is the leading cause of blindness worldwide. The medical treatment of DR entails tremendous costs. It becomes more costly when loss of productivity and quality of life are taken into account. The treatment of DR is important to improve the overall care of diabetic patients and to delay the progress of DR. Therefore, a strategic screening approach should be implemented in the early stage of diabetes mellitus diagnosis. Based on our study results, adults as young as 21years old were already diagnosed with T2DM in Sabah. In other words, DR does

not only affect patients with longstanding T2DM but also, in adolescent diabetes. Thus, diabetic eye screening is imperative once the patient is diagnosed with T2DM. The initial screening is important to assess for any pathological changes. Subsequent annual check-ups are required to evaluate the progression of DR. Annual screening of DR is also vital to provide a comprehensive guidance for physicians to adjust the diabetic management accordingly. While DR is not fully reversible, continuous provision of good diabetes care with intensive blood glucose control can delay its progression.

Lastly, aggressive diabetic management in the presence of nephropathy may slow the progression of DF and neovascular glaucoma. The importance of good blood sugar control is one of the key actions in preventing and delaying the onset of DR. If the Glomerular Filtration Rate (GFR) is slowly declining, a referral to a nephrologist should be considered. In addition, self-management is a crucial element of good diabetes care. Interventions that promote the adoption of healthy behaviours have been shown to significantly prevent or delay the onset of DR among T2DM patients with an increased risk of this disease. However, there are certain challenges faced by the Sabah community, including low literacy level, especially among the semi-rural and rural communities as well as limited English and Malay language literacy for them to understand the important concepts in T2DM management. Thus, the local level of poverty, knowledge on disease and cultural differences must be considered.

Among the T2DM patients with DR in Sabah, seven factors were strongly associated with DR, namely DFU, diabetic nephropathy, hypertension, dyslipidaemia, HbA1c, duration of diabetes mellitus (T2DM), and age of the patient. It shows the importance of regular diabetic eye screening and the need to optimize the care of all the preventable factors associated with DR. While diabetic control is a global issue, important variation at the local settings must be taken into consideration when tackling the known risk factors. Further studies can be performed to better examine the associations between these factors. Diabetes patients should receive a good quality of care from healthcare service providers. In addition, this study did not explore the association between geographical factors and DR. Therefore, future research should explore the impact of geographical factors towards the development of DR in Sabah. Based on the study results, necessary modifications in the management plan of DR can significantly improve the quality of life of diabetic patients.

## Acknowledgments

We would like to thank the Director General of Health Malaysia for his permission to publish this article. The authors would also like to express our gratitude to the Sabah State Health Department for their kind cooperation and support throughout the study.

## Author Contributions

**Conceptualization:** Nurul Athirah Naserrudin, Mohammad Saffree Jeffree, Nirmal Kaur, Syed Sharizman Syed Abdul Rahim, Mohd Yusof Ibrahim.

**Data curation:** Nurul Athirah Naserrudin, Mohammad Saffree Jeffree, Nirmal Kaur, Syed Sharizman Syed Abdul Rahim, Mohd Yusof Ibrahim.

**Formal analysis:** Nurul Athirah Naserrudin, Mohammad Saffree Jeffree, Nirmal Kaur, Syed Sharizman Syed Abdul Rahim, Mohd Yusof Ibrahim.

**Investigation:** Nurul Athirah Naserrudin.

**Methodology:** Nurul Athirah Naserrudin, Nirmal Kaur, Mohd Yusof Ibrahim.

**Supervision:** Mohammad Saffree Jeffree, Syed Sharizman Syed Abdul Rahim, Mohd Yusof Ibrahim.

**Validation:** Nurul Athirah Naserrudin, Mohammad Saffree Jeffree, Nirmal Kaur, Syed Sharizman Syed Abdul Rahim, Mohd Yusof Ibrahim.

**Visualization:** Nurul Athirah Naserrudin, Mohammad Saffree Jeffree, Nirmal Kaur, Syed Sharizman Syed Abdul Rahim, Mohd Yusof Ibrahim.

**Writing – original draft:** Nurul Athirah Naserrudin.

**Writing – review & editing:** Nurul Athirah Naserrudin, Mohammad Saffree Jeffree, Nirmal Kaur, Syed Sharizman Syed Abdul Rahim, Mohd Yusof Ibrahim.

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
