## [Decision Letter · Decision Letter 0]

11 Jun 2021

PONE-D-21-17347

Diabetic Retinopathy Among Type 2 Diabetes Mellitus Patients in Sabah Primary Health Clinics – The Underlying Factors

PLOS ONE

Dear Dr. Naserrudin,

Thank you for submitting your manuscript to PLOS ONE. After careful consideration, we feel that it has merit but does not fully meet PLOS ONE’s publication criteria as it currently stands. Therefore, we invite you to submit a revised version of the manuscript that addresses the points raised during the review process.

We look forward to receiving your revised manuscript.

Kind regards,

Kanhaiya Singh, Ph.D

Academic Editor

PLOS ONE

Journal Requirements:

Additional Editor Comments (if provided):

Reviewers have found this study interesting but they have recommended more detailed literature discussion. In addition, specific methodological details are required.

Please make all the supplemental information available to the reviewers.

Reviewers' comments:

Reviewer's Responses to Questions

**Comments to the Author**

1. Is the manuscript technically sound, and do the data support the conclusions?

Reviewer #1: Yes

Reviewer #2: Yes

2. Has the statistical analysis been performed appropriately and rigorously? 

Reviewer #1: Yes

Reviewer #2: Yes

3. Have the authors made all data underlying the findings in their manuscript fully available?

Reviewer #1: Yes

Reviewer #2: No

4. Is the manuscript presented in an intelligible fashion and written in standard English?

Reviewer #1: Yes

Reviewer #2: Yes

5. Review Comments to the Author

Reviewer #1: In this paper the authors are examining the prevalence of Diabetic Retinopathy (DR) in the state of Sabah, Malaysia and perform a cross-sectional study on 22,345 patients with Type 2 Diabetes Mellitus (DM), to identify the factors influencing the development of DR.

Overall, the findings of this manuscript are well-supported by the data and the methods used are appropriate. However, there are some points to consider that I have outlined below:

1. The authors mention in the disclosure that ‘All relevant data are within the manuscript and it’s Supporting Information files’. Are there any supporting information files? If yes, I don’t see them.

2. Lines 97-103, the authors talk about the unique characteristics of the state of Sabah and how it may cause a difference in the prevalence of DR. However, this has not been discussed adequately throughout the rest of the manuscript. How do the findings support the low prevalence of DR in Sabah? What is the dependence of DR on these external factors? How can one address this issue?

3. There are a couple of other papers on the prevalence of DR in other regions of Malaysia that have not been addressed/included in the manuscript. I have included some of the article links below, however, the authors should look into more relevant papers and include them in the discussion of this manuscript.

https://www.ncbi.nlm.nih.gov/pmc/articles/PMC7590874/

https://pubmed.ncbi.nlm.nih.gov/25470640/

4. The authors lightly touch upon proliferative and non-proliferative DR. Can the authors explain and address this in more detail? Also, what is the relevance of the type of DR with the main findings of this manuscript?

5. In Table 2, please define all the abbreviations in the column headings. Also, please write the full form of DFU in the table, so that the reader does not have to reference the text to understand the table data.

6. Line 100-102, what is the link between inadequate healthcare options for people of Sabah and the prevalence of DR?

7. In the methods, please have different sub-sections.

8. Tables 1 and 2, the authors may want to consider adding column borders to improve the clarity of the data.

9. In some places, the authors mention ‘ischaemic’, in other places it is ‘ischemic’. Please make it consistent.

10. Lines 86-90, are very repetitive with the previous paragraphs. It seems redundant. The authors should concise the information presented.

11. Line 191 and 206, have spellings as HBA1c and HbA1c, respectively. Please make the capitalization of letters consistent throughout the manuscript.

12. Throughout the manuscript, there are inconsistencies with the abbreviation of Diabetes Mellitus. In some places it is mentioned as DM, in others T2DM. Kindly correct it.

Minor concerns:

1. Line 47, should be ‘this study attempts to…’

2. Line 48, should be ‘factors influencing DR…’

3. Line 54, should be ‘By understanding these underlying factors…’

4. Line 56, should be ‘this study could also become the benchmark…’

5. Line 62, should be ‘global incidence of DM in 2017 was as high as…’

6. Line 77, should be ’24 and 74 years old…’

7. Line 83, the word diagnosed is used twice in the same sentence. Please rephrase.

8. Line 85, should be ‘develop some degree of…’

9. Line 93, requires a citation.

10. Line 104, it would be better to rephrase.

11. Line 143, should be ‘was 1.1 times higher than females…’

12. Line 158, should be ‘risk factor were patients with a DFU…’

13. Line 158, should be ‘was 3 times higher than…’, in order to maintain consistency throughout the manuscript.

14. Line 160, should be ‘was 2 times higher than…’, in order to maintain consistency throughout the manuscript.

15. Line 169, kindly rephrase to ‘from the findings of this study…’

16. Line 198, should be ‘other complications of diabetes…’

17. Line 229, needs a citation.

18. Line 242-243, should be one sentence instead of two.

19. Line 257-258, please make it concise, there are two ‘and’ in the sentence, which makes it redundant.

20. Line 268, should be ‘Acknowledgements’.

Reviewer #2: Ref:PONE-D-21-17347:

In the present article entitled “Diabetic Retinopathy Among Type 2 Diabetes Mellitus Patients in Sabah Primary

Health Clinics – The Underlying Factors” Naserrudin et al. have explored the RFs involved in DR. The study design is good and the results are well supported by the data. However, some points need to be addressed to make the study robust for the publication. Let's explore some of them from a clinical perspective

Major:

Supplementary sheet to support data

Mention about latest possibilities like pancreas transplantation and islet cell transplantation as an intervention (with latest references). It is expensive and puts financial burden on the community, so more important is good dietary habits and active lifestyle on a community level.

Minor:

Title is clear. “ The Underlying Factors” needs to be rephrased to something like 'addressing the underlying factors'.

Abstract:

Conclusion.

Page 2, line 37 “The effective management of the risk factors is crucial in preventing DR” can be changed to 'effective glycemic control.'

Main text:

line 83: change 'diagnosis' to 'disease progression'

142: "the probability kindly re-phrase to 'The odds ratio was..'

Overall need to address the latent phase of non-communicable disease like DM, HTN and modern interventions

Good luck,

6. PLOS authors have the option to publish the peer review history of their article (what does this mean?). If published, this will include your full peer review and any attached files.

Reviewer #1: No

Reviewer #2: **Yes: **Tejas Nikumbh

---

## [Author Response · Author response to Decision Letter 0]

4 Nov 2021

(i) Reviewer One

Thank you for the comments and suggestions to improve the manuscript. All the comments and suggestions have been carefully addressed by the author and co-authors. We have tried our best to follow all the suggestions and corrected all the typo as suggested by Reviewer One. We also have included recent Malaysian Studies as pointed out by Reviewer One. A point by point rebuttals have been prepared in the Response to Reviewers's file for your kind perusal. Thank you

(ii) Reviewer Two

Thank you for the comments and suggestions to improve the manuscript. All the comments and suggestions have been carefully addressed by the author and co-authors. We have included recent intervention for Diabetic Retinopathy as suggested by Reviewer Two. A point by point rebuttals have been prepared in the Response to Reviewers's file for your kind perusal. Thank you.

---

## [Decision Letter · Decision Letter 1]

29 Nov 2021

Diabetic Retinopathy Among Type 2 Diabetes Mellitus Patients in Sabah Primary Health Clinics – The Underlying Factors

PONE-D-21-17347R1

Dear Dr. Naserrudin,

We’re pleased to inform you that your manuscript has been judged scientifically suitable for publication and will be formally accepted for publication once it meets all outstanding technical requirements.

Kind regards,

Kanhaiya Singh, Ph.D

Academic Editor

PLOS ONE

Additional Editor Comments (optional):

Please address these minor errors as suggested by Reviewer 1 during proof reading stage.

The authors have addressed all the revisions and the manuscript looks complete. There a couple of minor errors. Please see below: Line 158, please correct ‘T2T2DM’ Line 159, please correct ‘patients with type 1 T2DM’ Line 194, do the authors mean ‘odds ratio’? It currently reads ‘odd ratios’.

Reviewers' comments:

Reviewer's Responses to Questions

**Comments to the Author**

1. If the authors have adequately addressed your comments raised in a previous round of review and you feel that this manuscript is now acceptable for publication, you may indicate that here to bypass the “Comments to the Author” section, enter your conflict of interest statement in the “Confidential to Editor” section, and submit your "Accept" recommendation.

Reviewer #1: All comments have been addressed

Reviewer #2: All comments have been addressed

2. Is the manuscript technically sound, and do the data support the conclusions?

Reviewer #1: Yes

Reviewer #2: Yes

3. Has the statistical analysis been performed appropriately and rigorously? 

Reviewer #1: Yes

Reviewer #2: N/A

4. Have the authors made all data underlying the findings in their manuscript fully available?

Reviewer #1: Yes

Reviewer #2: Yes

5. Is the manuscript presented in an intelligible fashion and written in standard English?

Reviewer #1: Yes

Reviewer #2: Yes

6. Review Comments to the Author

Reviewer #1: The authors have addressed all the revisions and the manuscript looks complete. There a couple of minor errors. Please see below:

Line 158, please correct ‘T2T2DM’

Line 159, please correct ‘patients with type 1 T2DM’

Line 194, do the authors mean ‘odds ratio’? It currently reads ‘odd ratios’.

Reviewer #2: (No Response)

7. PLOS authors have the option to publish the peer review history of their article (what does this mean?). If published, this will include your full peer review and any attached files.

Reviewer #1: No

Reviewer #2: No

---

## [Editor Report · Acceptance letter]

3 Dec 2021

PONE-D-21-17347R1 

Diabetic Retinopathy Among Type 2 Diabetes Mellitus Patients in Sabah Primary Health Clinics – *Addressing the Underlying Factors*

Dear Dr. Naserrudin:

I'm pleased to inform you that your manuscript has been deemed suitable for publication in PLOS ONE. Congratulations! Your manuscript is now with our production department. 

Kind regards, 

on behalf of

Dr. Kanhaiya Singh 

Academic Editor

PLOS ONE